# A Life Cycle Assessment Study of the Impacts of Pig Breeding on the Environmental Sustainability of Pig Production [note 1]

**DOI:** 10.3390/ani14162435

**Published:** 2024-08-22

**Authors:** Greg J. Thoma, Banks Baker, Pieter W. Knap

**Affiliations:** 1Resilience Services PLLC, 1282 S. Sherman Street, Denver, CO 80210, USA; 2Genus-PIC, 100 Bluegrass Commons Boulevard, Hendersonville, TN 37075, USA; banks.baker@genusplc.com; 3Genus-PIC, Lorbeerrosenweg 10, 30916 Isernhagen, Germany

**Keywords:** LCA, animal breeding, pig, environmental impact, climate change, global warming

## Abstract

**Simple Summary:**

Pig production has environmental impacts such as global warming and eutrophication. This is closely linked to the feed efficiency of production, i.e., how much feed is required to produce a given quantity of pig meat; this depends on traits such as growth rate, feed intake, and litter size. Commercial pig breeding creates genetic improvement in those traits, and this should reduce the environmental impact. We applied two life cycle assessment (LCA) studies to quantify this reduction. In the first LCA, we compared the environmental impact of pigs in 2021 to the predicted impact for 2030, and we found a 7–9% improvement over those 9 years. In the second LCA, we compared the impact of pigs from a particular breeding company to the North American pig industry average, and we found that those pigs have a 7–8% lower impact. We conclude that commercial pig breeding delivers positive environmental outcomes as a result of its selection for production and reproduction traits.

**Abstract:**

Lifecycle assessment (LCA) quantified changes in environmental impact categories (global warming, eutrophication, etc.) from 2021 to 2030 due to genetic trends in (re)production traits in pig lines of the breeding company Genus-PIC. The 2030 levels were projected with selection index theory based on weightings of traits in the breeding goals and genetic covariances among them. The projected improvement was 0.9% annually for most impact categories. Another LCA compared the impacts of 2021 North American pig production based on PIC genetics versus the industry average. Software openLCA converted material and energy flows to impact categories of frameworks ReCiPe-2016, PEF-3.1, and IPCC-2021. Flows came from data recorded by customers (1.1/4.7 million sows/finishing pigs) and by subscribers to a third-party data aggregator (1.3/9.1 million). PIC genetics have a 7–8% better impact than industry average for 13/18 categories of ReCiPe-2016, 19/25 of PEF-3.1, and all categories of IPCC-2001. Pig breeding delivers positive environmental outcomes as correlated responses to selection for profitability-oriented breeding goals. This trend is additive; technology development will increase it. Different investment levels in breeding population structure and technology and different operational efficiencies of breeding companies cause substantial differences in the environmental impact of pig production.

## 1. Introduction

Pig production has long been guided by efforts to increase productivity and decrease production costs; this also leads to an indirect reduction in environmental impacts. Consumers and businesses are increasingly aware of sustainability issues in our food systems, which have increased the pressure for producers to become more efficient, more focused on environmental impact, and more strategic about resource allocation and investment. The livestock sector uses resources such as land, water, and energy to produce animal products, and these activities impact air, water, and soil quality [1]. An important component of sustainability is supported by this focus on resource efficiency, so that the global availability of high-quality, affordable, and safe animal protein can be increased and its overall environmental impact can be reduced at the same time. This challenge is intensified by global climate change and environmental degradation, by the risks to food security that follow from these issues, and by the increasing demand for animal-sourced protein by the growing global middle class.

The challenges of sustainability go beyond regulatory compliance and include identifying innovations to increase production efficiency, safety, and resilience. Sustainability is good for business because it addresses risks and creates opportunities to increase efficiency and decrease negative impacts [2]. There remain gaps in our understanding of the impacts of production systems and in our ability to make informed decisions to guide changes in breeding, feeding, waste management, and other practices. A better understanding of these systems and their interactions will support the development and implementation of technology and management practices to maintain or increase productivity and improve sustainability at the same time. System-scale models integrate scientific knowledge, management practices, and the environmental and economic impacts of agroecosystems to support a rational basis for decision making.

The basic principle of life cycle assessment (LCA) is to follow a product through its life cycle by defining a boundary between its production system and the surrounding environment and tracking the inputs and outputs of all the activities involved. LCA is commonly used in sustainability studies that quantify the potential environmental impacts of agricultural production systems and the tradeoffs between them, providing an overall framework for measuring sustainability characteristics. A brief overview of the methodology from the point of view of livestock production is given by [3].

The environmental impact of pig production depends on many factors; the dominant factors are feed production and its utilization (feed conversion) and manure management. The animal as such plays a pivotal role here because its intrinsic capacities for lean tissue growth, feed efficiency, survivability, and reproductive performance influence these factors. Those performance characteristics are partly under genetic control. Livestock breeding increases farm animal productivity so that more animal protein can be produced by fewer animals, using less land, water, and other resources; it also increases the efficiency of that production, which reduces the emission intensity of undesirable outputs such as greenhouse gases. An example of a focus on sustainability specifically related to livestock genetics and breeding is the Code EFABAR initiative (http://www.responsiblebreeding.eu, accessed on 11 July 2024), where breeding and sustainability intersect multiple metrics: animal health and welfare, environment, resource use efficiency, and biodiversity.

It would then be relevant to quantify the contribution of genetic improvement to positive environmental and sustainable outcomes for the pig meat production system. In this study, we first present the genetic trends of the abovementioned traits (lean tissue growth, feed efficiency, survivability, and reproductive performance) as they were realized in PIC’s pig breeding populations from 2010 to 2023. We then apply life cycle assessment to transform such trends into trends of environmental impact factors, and we present a forecast to 2030. Finally, we compare the environmental impacts of PIC’s populations to the North American industry average of 2021 (a baseline comparison).

## 2. Material and Methods, or: Goal and Scope

Feed production causes much of the greenhouse gas (GHG) emissions from pig production; the 84 studies cited in the first paragraph of Section 4.2 report proportions between 22 and 95% (59 ± 18%). In line with this, the GHG emissions from a pig farm are strongly correlated to its whole-enterprise feed conversion ratio (FCR_we_); FCR_we_ can be quantified in terms of feed intake, growth rate, and survival rate of the growing–finishing pig, as well as feed intake and the lifetime reproductive performance of the sow. Appendix A describes the associated equations. Commercial pig breeding creates genetic change in all these traits, so the contribution of genetic change to FCR_we_ can be quantified, and from there, the change in GHG emissions due to genetic improvement can be predicted. Section 2.4 describes how we quantify this.

### 2.1. Functional Unit, System Boundary, Cutoff Criteria and Multi-Functionality

The functional unit of this study is the delivery of 1000 kg pig live weight at the farm gate; this includes culled sows from the upstream breeding herd. It does not include any animals not fit for slaughter; these do not enter the food chain (no live weight delivered, no desirable output), but their resource consumption, their emissions, and other undesirable outputs are included in the calculations. The systems being compared are (i) the average slaughter pig produced from the North American breeding program of a particular breeding company (Genus-PIC of Hendersonville, TN, USA; see www.pic.com) versus (ii) North American industry average genetics, assuming identical animal husbandry practices, farm infrastructure, post-farm activities, and ancillary activities such as accounting and travel. Our system boundary (Figure 1) begins with the extraction of raw materials and includes all animal husbandry activities, feed production (production of fertilizer, pest and disease control chemicals, seed, fuel, and energy), manure management, and veterinary and genetic research and development (R&D) activities.

No simplifying cutoff criteria [4] are implemented.

The primary multi-functional processes are the parent, grandparent (GP), and great-grandparent (GGP) stages (collectively known as the genetic pyramid; the “parent” is the parent of the slaughter pig) from where culled sows (a coproduct) are sent to slaughter; all environmental impact from these stages is assigned to the progeny, and an internal credit for the displaced growing–finishing pigs is assigned to the GP and GGP sow barns (similar to a consequential LCA approach to avoid allocation). Likewise, male piglets produced in the GP stage form another coproduct; they are sent to finishing, where they displace piglets from the parent sow barn. Thus, the live weight at the farmgate accounts for all the animals leaving the system to slaughter (Figure 1).

An inventory of inputs and emissions is created for each activity and constructs the cradle-to-farmgate lifecycle inventory (LCI) model, linked from the GGP stage to the finishing stage.

### 2.2. Unit Processes and Their Data Requirements

The foreground system includes the main activities that deliver the functional unit of Section 2.1, i.e., a typical North American commercial pig production operation from gestation and farrowing facilities to finishing and slaughter. To simulate this operation and estimate the utility consumption, we use the Pig Production Environmental Calculator of the University of Arkansas Resiliency Center [5]. The primary data required are listed in Appendix A. We use the market group process for USA electricity (Ecoinvent V 3.9.1 cutoff system model) and the I/O model by Carnegie Mellon University [6] for veterinary and genetic R&D services. Environmental impacts associated with infrastructure (buildings, machinery, etc.) are not included for foreground processes except when an existing Ecoinvent (see the next paragraph) unit process includes infrastructure. Emissions associated with manure management (ammonia, methane, non-methane volatile organic compounds, and nitrous oxide) are not commonly measured and are therefore modeled. This requires estimates of manure excretion and composition; we follow the LEAP recommendations (https://www.fao.org/partnerships/leap/en, accessed on 11 July 2024) to treat manure as a residual, consistent with the treatment of similar products in the background. The most relevant data gap in this study is the industry-wide distribution of systems for manure management and subsequent application; we model a uniform use of deep pit systems, as used in a significant fraction of North American production systems.

The background system includes all upstream activities. Purchased inputs to the main production processes (electricity, fuel, transportation, etc.) are sourced from the Ecoinvent v3.9.1 cutoff database ([7]; https://ecoinvent.org/the-ecoinvent-association, accessed on 11 July 2024); some feed ingredients are sourced from the Agrifootprint 5.0 database [8,9]. In North America, activities downstream of the abattoir are not affected by genetic factors, so we exclude them from the system boundary.

### 2.3. Realized Genetic Trends in Performance Traits

The genetic trend of a trait is commonly calculated by relating the trait’s estimated breeding values (EBVs) of a group of animals to the animals’ date of birth. In this case, the group included animals born and performance tested in PIC’s genetic nucleus farms during the past 11 years, of the four genetic lines behind PIC’s main types of slaughter pigs. The EBVs are routinely calculated, within genetic lines and on a weekly basis for all animals with a record in PIC’s worldwide database. The methodology follows [10], extended with the use of genomic information following [11]; this same methodology is in use by all serious livestock breeding organizations worldwide. For this LCA study, we focused on the traits that directly influence a farm’s whole-enterprise feed conversion ratio (FCR_we_, i.e., the amount of feed required to produce the functional unit of Section 2.1): the sow’s litter size (number born alive) and the lactation mortality rate of her piglets; and the growing–finishing pig’s feed intake, growth rate, and wean-to-finish mortality rate.

### 2.4. Scenario 1: Forecasting to 2030

For this forecast, we assume that the composition of the various pig diets remains constant over time and that the gross sustainability of feed production is not changed due to progress in animal genetics or management.

The genetic improvement in PIC’s routine selection traits is transformed into genetic improvement in the environmental impact factors of the LCIA. This is a semi-empirical process based on the currently active breeding goals of the four genetic lines behind PIC’s main types of slaughter pig. Each line’s breeding goal includes about 30 traits. For each animal in the breeding program, the routine EBVs (see Section 2.3) for each trait are weighted into a selection index. Selection of the next generation of breeding pigs on that index will result in genetic change (ΔG) in the underlying traits. This change can be predicted from the regression coefficients of the trait EBVs on the index; in vector form,
$$\Delta G=i\times \frac{b^{\prime }E}{\sqrt{b^{\prime }Eb}}$$
where *i* is the selection intensity on the index, b is the vector of index weighting factors, and E is the covariance matrix among the trait EBVs [12]. The weighting factors are constants that quantify the breeding goal, set to reflect each trait’s economic impact on farm profitability; for the forecasting in this study, E held the covariances among the trait EBVs of animals of each genetic line that were born in PIC’s nucleus farms the previous 18 months. In that same period, the genetic improvement in the index was 20.2 points per year [13] (p. 14), which is close to one standard deviation, so *i* can be set to 1.

As mentioned in Section 2.3, the combined genetic trends of some of these traits lead to an annual change in FCR_we_; refs. [14,15,16,17,18,19] have shown that FCR is strongly correlated with nitrogen excretion, which is the main contributor to the GHG emissions from monogastric animals. We may then expect a profitability-oriented pig breeding program to result in a correlated genetic reduction of those emissions: “mitigation as a co-benefit to improved production efficiency” [20]. Such effects can also be expected for other environmental impact factors such as eutrophication, acidification, and land use.

### 2.5. Scenario 2: Baseline Comparison

We model a generic pig production system with a sow farm (gestation and lactation) followed by a wean-to-finish farm with 5000 pigs (50:50 females and castrated males), with finished animals transferred to slaughter as the functional unit.

#### 2.5.1. Data

The baseline comparison data provide the same key performance indicators (KPIs: feed conversion, mortality, etc.) from two sources.

First, PIC customers across North America with a wide range of production and nutrition practices and systems, health statuses, and PIC crosses report performance records directly to PIC. This includes data recorded in 2021 on 1.1 million parent sows and 4.7 million growing–finishing pigs from 53 and 34 multi-farm production companies, respectively.

Second, North American industry average production data are provided in aggregated form by a prominent industry benchmark with similar data quality control protocols as used by PIC. This dataset represents a similarly wide range of production practices, nutrition programs, and health statuses. The aggregator does not differentiate data collection based on genetic products, and we believe the (undocumented) contribution from PIC genetics to this data to be between 30 and 50%.

Table 1 summarizes the data available from both sources. These sow performance data were recorded on crossbred parent sows; to properly cover the GGP and GP stages in the model (i.e., purebred sows producing purebred and crossbred litters, respectively), their reproduction traits were adjusted for maternal and direct heterosis based on [21] (Table 10.2). Similarly, growth rate and feed efficiency in gilt rearing at these stages were heterosis-adjusted based on [22] (p. 347). Costs of USD 3.5 per finished pig and USD 2.5 per piglet were adopted based on enterprise production budgets (https://www.extension.iastate.edu/agdm/decisiontools.html#livestock, accessed on 11 July 2024). Veterinary and genetic R&D costs were estimated from financial data from [23] coupled with estimates of the genetic pyramid herd size; this led to an expenditure of about 22,000 USD per GGP sow. Emissions were found to be insensitive to this parameter; therefore, any uncertainty in this estimate will not affect the study conclusions.

#### 2.5.2. Diet Composition

Diets for sows and for growing–finishing pigs were formulated according to the nutrition and feeding guidelines of [24], based on ingredients typically used in North America.

The amounts of sow feed consumed in the gestation and lactation phases were estimated with PIC’s Dynamic Sow Feeding Tool (http://dynamicfemalefeeding.pic.com/en, accessed on 11 July 2024). For the industry average evaluation, the same diets and computational procedures were used, averaging the feeding guidelines of PIC and of other breeding companies in North America because the industry average data does not specify the genetic background of its pigs. Appendix A gives the diet compositions.

For growing–finishing pigs, diet composition was derived from the growth rate and feed conversion performance reported in the baseline comparison data of Section 2.5.1. The allocation per phase was determined by the weaning body weight, market body weight, and overall feed conversion using the PIC Feed Budgeting tool (https://www.pic.com/resources/nutrition-links-and-tools, accessed on 11 July 2024; overall growth rate 0.77 kg/d, overall FCR 2.55 kg/kg). Similar procedures were used for the industry average to adjust for the differences in growth rate and feed conversion (0.75 kg/d, 2.71 kg/kg). Appendix A gives the diet compositions.

### 2.6. Lifecycle Impact Assessment

LCIA frameworks consist of a series of category-specific characterization factors used to convert the cumulative inventory results of an LCI model to the cumulative impact in multiple environmentally relevant categories for the functional unit of the system under study. For example, the climate change impact is calculated by taking the product of each greenhouse gas emission and its respective characterization factor to convert the cumulative, relative radiative forcing into carbon dioxide equivalents, CO_2_ eq. We use three LCIA frameworks to characterize the environmental impact of the functional unit, as follows.

*ReCiPe-2016* includes 18 impact categories covering the areas of protection identified under human health, ecosystem health, and resource conservation. It includes a 2010 world-based normalization method that can aid in defining which categories are relatively larger in the context of the annual average global per capita emissions. We apply all three of the cultural perspectives of ReCiPe 2016: egalitarian, hierarchist, and individualist.

*Environmental Footprint-3.1* is the framework required for the Product Environmental Footprint (PEF) recommended by the European Commission. It includes 25 impact categories, 14 of which are qualitatively the same as the ReCiPe 2016 categories but may have different quantitative reporting units.

*IPCC-2021 climate change* includes twelve climate-related impact categories. Its category of primary interest for this study is the 100-year global warming potential (“GWP-100”), as it coincides with the climate change categories in ReCiPe and PEF. Other categories include global warming potential for different time horizons and global temperature change potential.

The lifecycle impact assessment (LCIA) calculations are performed using openLCA v2.02 to link the individual stages of production, creating a supply chain model to convert the material and energy flows of pig production to the impact categories of the three frameworks. Background data are separately licensed from the Ecoinvent v3.9.1 cutoff and APOS system models and the Agri-footprint 5.0 databases; this includes the probability distributions for emissions of ammonia, nitrous oxide, etc. from crop production. We adopt the background database’s uncertainty distributions without revision.

### 2.7. Contribution Analysis

The contributions of the various processes to each impact category are quantified for scenario 1 (Section 2.4). The inventory projection focuses only on the KPIs of the production systems; it does not include projected changes in efficiency or other contributing sectors such as electricity, crop yield, or transportation. Hence, the contributions for scenario 2 (Section 2.5) are very similar to those of scenario 1, and we do not perform a separate contribution analysis.

### 2.8. Methodological Quality Assessment

LCA has several sources of uncertainty that can influence the interpretation of the results and limit the conclusions [25,26]. These include (i) natural variability in input parameters such as fertilizer or fuel use, (ii) estimated values obtained through proxy sources, substituting a similar product for one that does not exist in available databases, and (iii) results from mathematical models that include multi-year simulations of parameters associated with variable factors such as weather or soil conditions. There are also uncertainties in impact assessment characterization factors, but the studied systems are quite similar; therefore, we expect these to be uniform across all compared systems and not affect the study conclusions. Nonetheless, the selection of an LCIA framework is a value choice and should be tested for sensitivity. Comparison of the results from the ReCiPe and the PEF frameworks (Section 2.6) evaluates the robustness of the results.

We quantify the uncertainty in the LCA results using Monte Carlo simulation (MCS), available in the openLCA software. This process is rules based and incorporates prior probability distributions for uncertain or variable input data that reflect the knowledge and process uncertainty associated with the variables and describe the range of expected or permissible parameter values. We apply log-normal prior probability distributions to avoid sampling of negative input values—most of the reference flows in LCA are positive definite, so negative values are not permissible. MCS produces posterior probability distributions for its predicted output variables (i.e., the impact categories), which can be used for statistical significance testing. We use the GLM procedure of SAS 9.4 [27] to perform a two-sided unpaired ANOVA on sets of 100 MCS results for each impact category to test the significance of the contrasts (i) between PIC-2021 and the 2021 industry average and (ii) between PIC-2021 and PIC-2030. We use the SAS procedure UNIVARIATE to test for normality of the data distribution.

Characterization factors are used to translate the inventory to the impact categories. The magnitude of an observed contrast is not necessarily indicative of its statistical significance: a factor of two for climate change may be significant, while a factor of 1000 for toxicity effects may not, partially due to the much larger range in characterization factors for toxicity. An additional explanatory factor arises in the process of uncertainty propagation in the MCS when the uncertainty of some of the contributing flows is higher, resulting in a wider frequency distribution in the MCS and therefore less differentiating power in the statistical tests. The flow uncertainty for foreground processes is quantified via the Ecoinvent pedigree matrix: Five characteristics of the data (*reliability*, i.e., how was it measured; *completeness*; *temporal correlation*, i.e., how recent is it; *geographical correlation*, i.e., where is it from; *technological correlation*, i.e., which technology generated it) are qualitatively assessed in terms of five *indicator scores*, scored from 1 for the highest quality, e.g., “verified data based on measurements”, to 5 for the lowest quality, e.g., “non-qualified estimate” [28] (Table 10.4). These are converted into five default variance values [28] (Table 10.5), which increase from zero to higher values for more unfavorable indicator scores. Each of these is combined with its corresponding *basic uncertainty* (estimated from aggregate statistics for the KPIs). A default variance value (based on expert judgement) is applied when no sampled data are available. This defines the final prior variance of the contributing flow as it goes into the MCS. Appendix A gives the elements of our inventory with indicator scores different from (1 1 1 1 1).

Our uncertainty assessment does not include uncertainty associated with the characterization factors. Because the systems being compared are very similar, this omission of a source of variability in the results will not introduce bias in the conclusions; any single MCS would have selected the same characterization factor for both systems.

## 3. Results

### 3.1. Realized Genetic Trends of Performance Traits

Realized genetic trends of the FCR_we_-relevant traits (see Section 2.3) are shown in Figure 2. The white trendlines in each graph represent the pureline breeding populations; the black trendline is the average of the white ones and hence represents the typical PIC-derived slaughter pig or parent sow. The pre-2022 time trends are very stable (for feed intake since 2017), and we predict they will continue into the future; this would lead to a forecast such as Scenario 1, which is reported in Section 3.2.1.

### 3.2. Scenario Analysis

For both scenarios studied here (Section 2.3 and Section 2.4), the estimates of the impact categories are presented not as absolute values but proportionally: for the forecast of Section 2.4, we report the PIC-2030 predicted values as proportions of the PIC-2021 results, and for the baseline comparison of Section 2.5, we report the PIC results as proportions of the industry average results. Fourteen of the eighteen and twenty-five impact categories of the ReCiPe-2016 and PEF-3 frameworks (see Section 2.6) are functionally similar across these frameworks, and this proportional approach makes them quantitatively comparable. For example, acidification is quantified in terms of SO_2_ eq (in kg) by ReCiPe-2016 but in terms of H^+^ eq (in mol) by PEF-3; this is resolved when both are transformed to a percentage of a similar associated reference value. In all these cases, a lower environmental impact is more favorable, but to stay conservative, we apply two-sided significance tests. Shapiro–Wilk tests confirmed the normality of the data distribution (*p* = 0.094 in one case, 0.1 < *p* < 0.995 for the rest). Levene tests for unequal group variances were significant (0.015 < *p* < 0.05) in 4% of the cases, and we report Welch variance-weighted *p* values there; in the most extreme case, this changed the *p* value from 0.01004 to 0.01013.

#### 3.2.1. Scenario 1: Forecast to 2030

As mentioned in Section 2.4, this forecast was not derived from an extrapolation of the historical trends in Figure 2, but from multivariate regression exploiting the covariance matrix among the trait EBVs, based on the currently active breeding goals of these breeding populations. The predicted 2030 levels of the performance traits are shown in Figure 2. The resulting projected 2030 performance of the LCIA environmental impact categories (Section 2.6) can be compared to the 2021 performance, and this suggests a predicted improvement of 7 to 9% (i.e., about 0.9% per year) for most of the impact categories in both the ReCiPe-2016 and PEF-3.1 frameworks; specifically, the predicted improvement of the global warming potential is 7.5% (i.e., 0.83% per year). The exceptions are for ionizing radiation (10.6% in ReCiPe-2016 and 12.3% in PEF-3.1); these are exceptions in the favorable direction, but the estimates are statistically non-significant (see the third paragraph of Section 2.8 for the paradox). All these results are presented in Appendix A. Figure 3 shows the results for the 14 functionally overlapping categories across the two frameworks.

#### 3.2.2. Contribution Analysis

Figure 4 presents the relative contribution analysis for the ReCiPe-2016 hierarchist and PEF-3.1 frameworks, both under the cutoff system model. The category axis has been sorted to emphasize commonalities across the contributing activities.

For ReCiPe-2016, the animal-rearing phases (finisher and piglet) are major contributors only to terrestrial acidification and fine particulate matter formation; both are dominated by ammonia emissions. Corn production is the dominant contributor across many impact categories; its primary contributing factor is emissions associated with fertilizer application. Metals, extracted and consumed in the supply chain, and fossil fuel consumption are the obvious dominant contributors to mineral resource and fossil fuel resource scarcity, respectively. The ionizing radiation category is dominated by waste management, with formaldehyde production as the primary contributing factor.

PEF-3.1 largely repeats the ReCiPe-2016 patterns, with interesting exceptions in the categories of ozone depletion, land use, water use (where corn production is the dominant contributor in ReCiPe but is very minor in PEF), and carcinogenic toxicity (where the opposite holds). In ReCiPe, land use is an indicator of potential impact on biodiversity associated with different land use classifications; in PEF, the main contributor is “other” processes, dominated by infrastructure such as buildings and roads, with some contribution from forestry. The primary “waste management” contributing factor to ionizing radiation in PEF-3.1 is polystyrene.

#### 3.2.3. Scenario 2: Baseline Comparison

Our comparison of the 2021 performance of PIC genetics to the industry average reveals that PIC genetics have a significantly lower environmental impact for 13 of the 18 impact categories in the ReCiPe-2016 framework, for 19 of the 25 categories in the PEF-3.1 framework, and for all twelve climate change categories in the IPCC-2001 framework. All these results are presented in Appendix A, and Figure 5 shows the results for the 14 categories that are functionally overlapping across ReCiPe-2016 and PEF-3.1 and for the GWP-100 category of IPCC-2021. For most of the impact categories, the PIC benchmark performance is 7 to 8% better than the industry average; because the industry average includes an undocumented proportion of PIC genetics (see Section 2.5.1 and Section 4.3), these estimates are conservative.

## 4. Discussion

### 4.1. Boundaries within the Current LCA Environment

This study is an environmental lifecycle assessment (an E-LCA); it focuses on the undesirable biophysical impacts of its functional unit. As a complementary activity, UNEP [29] formalized the socio-economic lifecycle assessment (the S-LCA), focusing on impacts on the well-being of stakeholders such as workers, consumers, and the local community, among others. UNEP [29] notices that such impacts may be desirable (for example, when they lead to employment opportunities or affordable food security) as well as undesirable. A complete lifecycle sustainability assessment of a production system would then involve an E-LCA, an S-LCA, and also a lifecycle costing exercise to quantify “the direct costs and benefits from economic activities”. Following [30], the *triple bottom line* of a production system must quantify its impacts on the three pillars of sustainability: *people*, *planet*, and *profit*. The latter pillar was euphemized to *prosperity* in the 2002 World Summit on Sustainable Development (https://www.un.org/en/conferences/environment/johannesburg2002, accessed on 11 July 2024); UNEP [29] adopted this terminology. S-LCA focuses on the *people* and *prosperity* pillars; E-LCA focuses on the *planet* pillar. As we argued previously [31] (p. 7974), a complete assessment of a livestock production system should involve a fourth pillar in its *quadruple bottom line*, focusing on the interests of the animals; for pig or poultry production, this would conveniently lead to a fourth P-labeled pillar, such as *pigs* (we could also think of gimmicks such as *Puminants* or *phish*). In terms of the present study, this would call for quantification of the impacts of pig breeding on pig welfare. Zira et al. [32] modeled this in an elegantly integrated way by introducing pigs as an additional stakeholder in the One Health approach [33] to assess the impacts of changes to (i) animal housing conditions and (ii) the breeding goal on (iii) the environment, on (iv) people, and on (v) pigs. A similar approach was followed by [34] in an S-LCA of broiler chicken production. Zira et al. [32] assessed the impact of (ii) on (v) in terms of piglet mortality rate, tail-biting incidence, and sow longevity. Unfortunately, changes to their items (i) and (ii) were confounded, so this study could not produce unequivocal conclusions in terms of (ii) breeding goals. Reflecting on his S-LCA-related work, Zira [35] (p. 86) concluded that this is “a daunting task if not herculean, requiring immense time. Trying to assess future livestock systems based on forecasts using scant current and historic data is also difficult”. Hence, “more research is required to develop an integrated assessment of the social, economic and environmental impacts of food production from farmed animals. Such an evaluation should also include the farm’s sensitivity to economic changes and competition for arable land for feed or food”, and “many methods of sustainability assessment have so far focused on negative environmental effects because they are the easiest to measure” [35] (p. 107). So, pending such “research to develop an integrated assessment”, we chose to restrict ourselves to the current E-LCA, noticing that measuring its effects credibly and defensibly is difficult enough.

### 4.2. Comparison to Other Studies

The scientific literature holds many LCA studies of livestock production systems; reviews of such studies include [36,37,38]. Pig-specific LCAs include the 74 studies cited by [39] plus, recently, [32,40,41,42,43,44,45,46,47,48,49]. Refs. [50,51] provide other reviews.

LCAs like our current study, specifically focused on the impact of livestock *breeding*, are much scarcer. We are aware of the following LCA studies that focus on pig breeding.

Refs. [43,52] conducted an LCA of the pig production sector in Norway. Based on the genetic improvement in feed efficiency, postweaning mortality, and sow reproductive output that was realized during the year 2021, the correlated genetic reduction of GHG emission intensity in that year was reported as 1.4% and 1.9% of the mean level for the Norwegian Landrace and Duroc populations, respectively. This mean level was 186 kg CO_2_ eq for a pig with 80 kg carcass weight, so the absolute annual reduction must have been 2.61–3.54 kg CO_2_ eq per slaughter pig.

Ref. [53] used the conversion factors described in our Appendix A to derive a genetic reduction of GHG emission from the genetic improvement of the traits described there, as realized between 2012 and 2023 in the same commercial pig crosses as covered in the current study. The correlated annual reduction was estimated at 2.93 kg CO_2_ eq per slaughter pig.

Alfonso [54] derived conversion factors for sow productivity traits (mainly litter size and piglet mortality rate) with respect to GHG emissions, similar to the ones described in Appendix A. As suggested by [55], he added these to the trait weighting factors of his profitability-oriented breeding goal, assuming a carbon emission shadow price of 0.04 EUR per kg CO_2_ eq. The impact of this addition was then quantified, and it was concluded “that no relevant changes are produced in the relative [weightings] of sow efficiency traits after the inclusion of GHG costs”. In other words, at that level of the carbon shadow price (which corresponds to the EU Emissions Trading System price of early 2021, see, e.g., https://www.investing.com/commodities/carbon-emissions, accessed on 11 July 2024), the inclusion of explicit GHG-related elements in a breeding goal that was designed without GHG emissions in mind does not add much to the predicted GHG output of the production system.

Alfonso’s [54] breeding goal did not include feed intake or feed efficiency traits, so his results are not surprising. By contrast, ref. [56] conducted a similar study with the same carbon shadow price but with a pig breeding goal based on litter size, piglet mortality, growth rate, and, notably, feed efficiency. One simulated generation of selection on this profitability-oriented breeding goal produced a correlated reduction of the GHG emission per slaughter pig of 0.510%; the addition of the carbon shadow price changed this to a reduction of 0.514%. So again, at that level of the carbon shadow price, an addition of explicit GHG-related elements to a profitability-oriented breeding goal (designed without GHG emission in mind but this time including feed efficiency) does not add much to the predicted GHG output of the production system.

Ottosen [57] (Chapter 5) calculated sets of trait weighting factors for breeding goals aimed at reducing the cost of pig production (i.e., profitability-oriented) or at reducing each of the ReCiPe-2016 impact categories (see Section 2.6 and our Figure 3). Each of these breeding goals was used in a deterministically simulated 10-generation pig breeding program with full specification of the genetic covariances among the traits, integrated into an LCA study to predict its impact on those same categories; in other words, the impact *on* each category was quantified *for* each category-oriented breeding goal, with the logical hypothesis that “a breeding goal designed to reduce one impact category will be substantially better at doing so with relatively smaller reductions in other impact categories”. However, their overall conclusion was that the various breeding goals “led to very similar reductions for each targeted impact category, with only minimal differences”. In more detail (derived from Figure 5.12c of [57]), these 10-generation (i.e., about 15 years) reductions ranged from 17.9–18.3% (for impact category fossil fuel depletion, across all the breeding goals) to 24.2–24.5% (for impact category acidification, across all the breeding goals): a 6.2% range across impact categories and a 0.3–0.4% range across breeding goals. Hence, “the present [profitability-oriented] breeding programme performs well in reducing all investigated environmental impacts”.

In summary, and consistent with our forecast of Section 3.2.1, current profitability-oriented pig breeding goals have been (and are still) reducing the GHG emissions of pig production by about 1% per year, dependent on local conditions and on the body weight trajectory studied.

### 4.3. Scenario 2: Baseline Comparison

Averaged across the LCIA frameworks in Figure 4, the global warming performance of PIC-2021 is 7.5% lower than the 2021 North American industry average level. Considering that the industry average data contain an undocumented proportion of PIC animals, which we believe to be between 30 and 50% (see Section 2.5.1), this result can be extrapolated to a 2021 global warming performance of PIC genetics that is 10 to 14% lower than that of the non-PIC-derived part of the North American pig sector.

Given the way our results were calculated, the lower environmental impact of PIC genetics as compared to the North American industry average is due to differences in production traits such as feed intake, growth rate, and survival rate of the growing–finishing pig, as well as feed intake and reproductive performance of the sow. Such differences between genetic populations are due to differences in genetic levels of the founder populations and in the subsequent rate of genetic change, which is determined by (i) the weighting of traits in the breeding goal and its associated selection index, (ii) the intensity of selection on that index at the GGP, GP, and parent levels, (iii) the accuracies of the estimated breeding values of the various traits, and (iv) the length of the generation interval at the GGP and GP levels (e.g., [58]). Most pig breeding companies have productivity- and/or profitability-oriented selection programs (e.g., [59]), so element (i) should not differ much between companies. Element (ii) is largely determined by population size, element (iii) by population size, by the quantity and quality of data recording, and by the quality of statistical data processing, and element (iv) by farm logistics. Generally, configurations with levels of elements (ii) to (iv) that lead to a faster rate of genetic change are more costly. Differences in the realized progress between companies will then be mostly due to different levels of investment in technology and population structure and to different operational efficiencies.

### 4.4. Extrapolation to 2030

Averaged across the LCIA frameworks in Figure 3, the predicted PIC-2030 global warming performance is 7.5% lower than the PIC-2021 level. An obvious question is then how the predicted PIC-2030 levels would compare to the predicted 2030 industry average values—not only for global warming but also for the other impact categories. The 2030 industry average values cannot be predicted with the semi-empirical approach of Section 2.4 because the selection index weighting factors and the genetic covariances of breeding programs other than PIC’s are not available to us. Instead, we fitted a logarithmic regression to the pre-2021 North American time trends of the relevant KPI traits (i.e., growth rate, feed intake, mortality rate, litter size, etc.) as reported by [60,61,62] and by AgriStats, and we extrapolated the fitted patterns to 2030. Background data and feed composition were kept unchanged. Appendix A gives the KPI data used and the projections.

Figure 6 shows the results for the 14 functionally overlapping categories across the ReCiPe and PEF frameworks; the predicted PIC-2030 levels are 13 to 16% more favorable than the predicted 2030 industry average values (14% for the global warming performance). Notice that this comparison carries a high level of uncertainty for two reasons.

First, the prediction methods are different: the PIC-2030 performance is based on semi-deterministic prediction making use of known trait covariances and known trait weighting factors, whereas the 2030 industry average was predicted by extrapolation of historical trends.

Second, future selection strategies (with their differential weighting of traits) are likely to differ from past (for the industry average) and current (for PIC) strategies. This may be in response to changing costs or prices or to externally imposed structural changes, e.g., in housing conditions, as recently triggered in the USA by California ballot proposition 12 (https://en.wikipedia.org/wiki/2018_California_Proposition_12#, accessed on 11 July 2024) and by similar trends in Europe (e.g., [63]). As we argued in Section 4.3, most pig breeding companies have productivity- and/or profitability-oriented selection programs, and similar changes in market or production pressure should lead to similar decisions around future selection directions. Future differences in the realized progress between companies will then be mostly due, again, to different levels of investment in technology and population structure and to different operational efficiencies. Also, novel technologies such as digital phenotyping (e.g., [64,65,66]), genome editing (e.g., [67]), and semen sexing may change the dynamics substantially over and above the conventional effects of selection.

In the LCIA, this increased uncertainty is accounted for using the Ecoinvent pedigree matrix, as described in Section 2.8.

### 4.5. Livestock Genetic Improvement

As we argued in Section 1, livestock breeding (i) increases farm animal productivity so that more animal protein (the desirable output) can be produced by fewer animals, using less land, water, and other resources; it also (ii) increases the efficiency of that production, which reduces the emission intensity of undesirable outputs such as greenhouse gases. Our results in Section 3.2.1 and Section 3.2.2 and the literature reviewed in Section 4.2 provide sufficient illustration of point (ii). With regard to point (i), Figure 7 shows how increased cattle and pig productivity has reduced the worldwide standing populations of these species, as they were required to cater for the production of a certain amount of animal protein. Across both species worldwide in 2020, this value was 46% of the 1960 level.

It follows that increased livestock productivity and efficiency have been reducing the footprint intensity of animal protein for many decades; this is fortunate, for the worldwide demand for (and, hopefully, the associated production of) animal-sourced food increased fourfold from 1960 to 2020 and has been projected to increase fivefold from 1960 to 2040 [68] (Figure 7). Hence, the current discussion about the livestock footprint would have been much more pessimistic if the intensity reduction had not taken place. Significantly, the contribution to this trend made by livestock breeding is cumulative and permanent, in contrast to interventions such as feed additives.

## 5. Conclusions

Current pig breeding programs have been delivering positive environmental outcomes as a correlated response to selection for the (re)production traits that feature in conventional profitability-oriented breeding goals. This trend will continue and is likely to become stronger due to technology development.

Our LCA study is the first to use long-term genetic trends in pig breeding and to forecast the mitigation of environmental impacts. The forecast predicts, for PIC-derived slaughter pigs in North America, a statistically significant genetic reduction in twelve impact factors, including the global warming potential, each by 0.7 to 1% per year up to 2030.

Such trends may also become stronger due to shifts in profitability-oriented breeding goals towards more focus on the traits that are most strongly related to environmental impact categories, but this will require considerably more focus than what current carbon shadow prices would dictate. This is largely a political issue.

Due to different levels of investment in technology and in the structure of breeding populations, and due to different operational efficiencies of North American pig breeding companies, the current (2022) status quo is that PIC genetics have a 7 to 8% lower impact on most environmental impact categories than the North American industry average.

## Figures and Tables

**Figure 1 animals-14-02435-f001:**
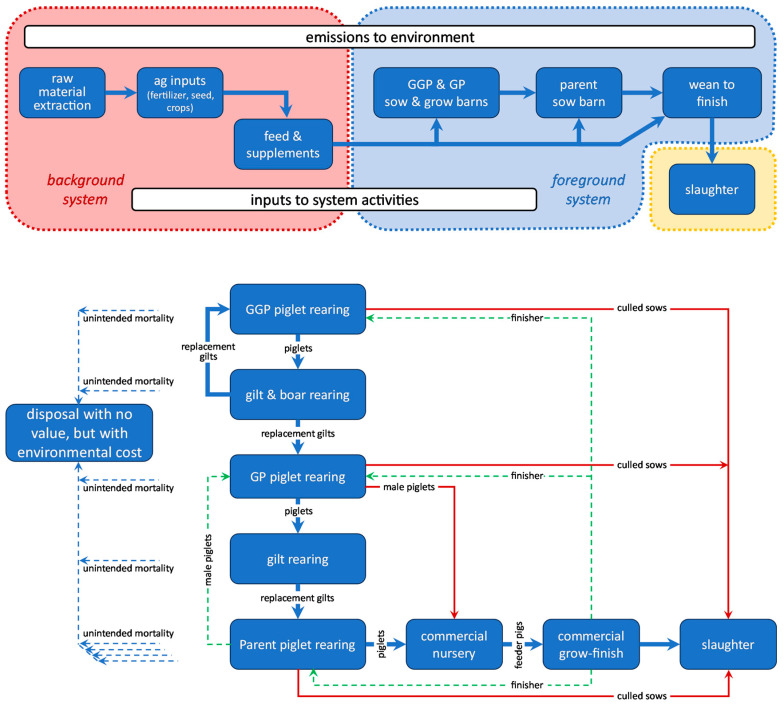
(**Top**): the cradle-to-farmgate system boundary. “GGP & GP” sows are the great-grandparents and grandparents of the slaughter pigs, respectively. (**Bottom**): the foreground system, showing animal flows (blue solid lines: the main product; red solid lines: coproducts; blue dashed lines: losses) and accounting flows for the allocation avoidance algorithm (green dashed lines).

**Figure 2 animals-14-02435-f002:**
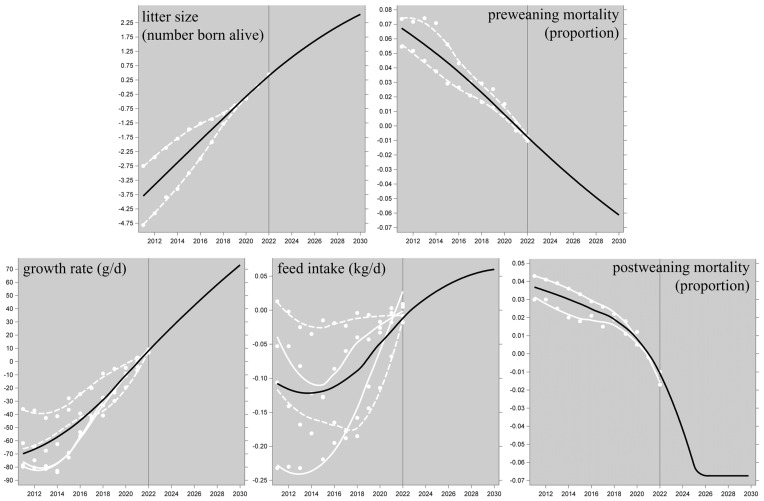
Realized genetic trends of the traits that influence a farm’s whole-enterprise feed conversion ratio FCR_we_. The white datapoints are annual within-line averages of the trait EBVs in two purebred sire lines (solid white trendlines) and two purebred dam lines (dashed white trendlines). Up to 2022 (vertical reference line), the black datapoints and trendlines represent the average realized performance of the resulting cross, i.e., the typical PIC-derived slaughter pig or parent sow. After 2022, the black trendlines represent the future performance as predicted by selection index theory (see Section 2.4); for postweaning mortality, this prediction reaches unrealistic values after 2025, and these were capped. The trendlines are LOESS interpolations. The EBVs on the y-axes are deviations from arbitrary means that were forced to zero for 2022.

**Figure 3 animals-14-02435-f003:**
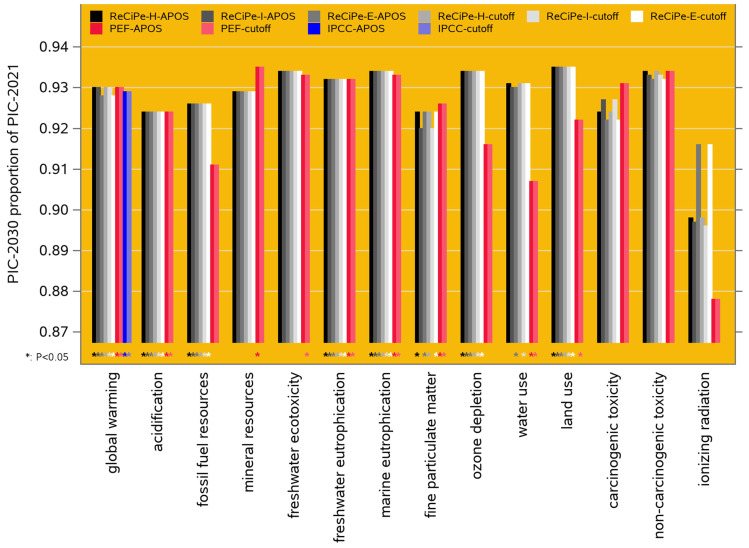
Results for scenario 1: predicted performance of PIC genetics in 2030 as a proportion of the 2021 performance for 14 environmental impact categories, as modeled in ten LCIA frameworks (ReCiPe-2016 egalitarian, hierarchist, and individualist, cutoff and APOS; PEF-3.1 cutoff and APOS; IPCC-2021 cutoff and APOS). The IPCC cases for global warming (blue bars) are IPCC’s GWP-100 category; IPCC does not consider the other impact categories. Lower values are more favorable; the asterisks below the bars indicate cases where the predicted 2030 proportions are significantly (*p* < 0.05) different from 1 (i.e., from the 2021 level).

**Figure 4 animals-14-02435-f004:**
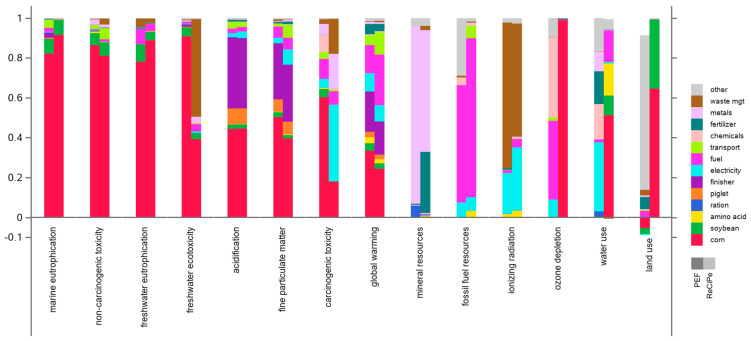
Contribution analysis for the ReCiPe-2016 hierarchist cutoff and PEF-3.1 cutoff LCIA frameworks. Note that the category axis is sorted differently than in Figure 3.

**Figure 5 animals-14-02435-f005:**
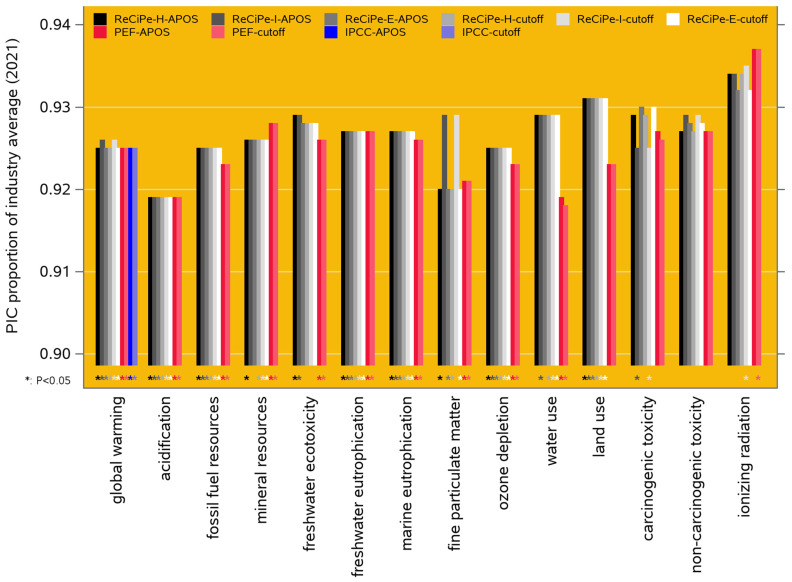
Results for scenario 2: performance of PIC genetics as a proportion of the North American industry average (both in 2021) for 14 environmental impact categories, as modeled in ten LCIA frameworks. Lower values are more favorable; the asterisks below the bars indicate cases where the PIC proportions are significantly (*p* < 0.05) different from 1 (i.e., from the industry average). The color coding and further formatting is the same as in Figure 3.

**Figure 6 animals-14-02435-f006:**
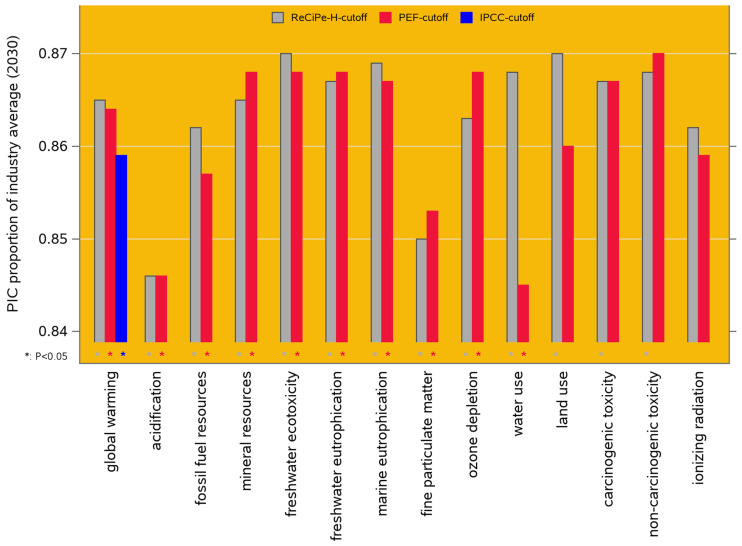
Predicted performance of PIC genetics in 2030 as a proportion of the predicted North American industry average in 2030. See Figure 3 for details.

**Figure 7 animals-14-02435-f007:**
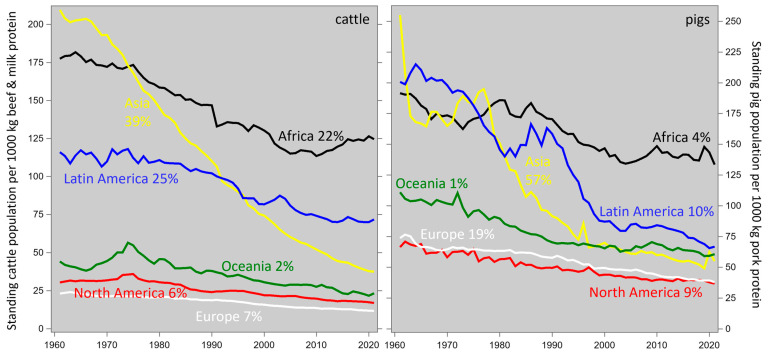
Time trends of the standing population size of cattle and pigs associated with the production of 1000 kg animal protein, by continent. The percentages in the continent labels represent the proportion of the 2020 global populations. Data from https://www.fao.org/faostat/en/#data, accessed on 11 July 2024; assuming 3.5% protein in milk and 16% protein in meat; note that this will underestimate the earlier y-values for beef and pork due to higher fat content, especially in Asia.

**Table 1 animals-14-02435-t001:** Performance KPI data from the PIC internal benchmark and the prominent industry benchmark, recorded in 2021. Means ± standard deviations.

	PIC-USA Internal Benchmark	North American Prominent Industry Benchmark
Sow performance		
Number of sows	1,093,952	1,273,698
Farrowing rate (%)	85.7 ± 4.12	91.6 ± 4.74
Total number born	15.2 ± 0.68	15.1 ± 0.62
Number born alive	13.7 ± 0.62	13.7 ± 0.53
Lactation mortality rate (%)	15.8 ± 3.69	16.2 ± 2.63
Weaning age (days)	21.4 ± 2.01	21.0 ± 1.42
Number weaned	11.6 ± 0.59	11.5 ± 0.62
PWMFY *	26.5 ± 4.17	27.0 ± 1.99
Sow mortality rate (%)	15.0 ± 4.70	14.1 ± 3.69
Wean-to-finish performance		
Number of pigs	4,741,133	9,134,940
Start weight (kg)	5.98 ± 0.59	5.53 ± 0.39
End weight (kg)	129.0 ± 5.31	129.1 ± 2.09
Days on feed	158.7 ± 14.7	167.1 ± 5.15
Growth rate (kg/d)	0.77 ± 0.060	0.74 ± 0.023
Feed intake (kg/d)	1.96 ± 0.15	1.94 ± 0.11
Feed conversion ratio	2.55 ± 0.17	2.63 ± 0.12
Mortality rate (%)	6.30 ± 4.40	9.88 ± 3.48

* PWMFY: piglets weaned per mated female per year.

## Data Availability

The datasets presented in this article are not readily available because they are commercially sensitive. Requests to access the datasets should be directed to pieter.knap@genusplc.com.

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
