# Peer review of "A Life Cycle Assessment Study of the Impacts of Pig Breeding on the Environmental Sustainability of Pig Production [Author-notes fn1-animals-14-02435]"

_animals, 2024, doi:10.3390/ani14162435_

Round 1

Reviewer 1 Report

Comments and Suggestions for Authors

4The manuscript submitted for review covers an interesting topic; however, the authors have not avoided some errors. The manuscript can be revised and accepted once corrections are made:

1.       lack of access to supplementary materials hinders full and proper review (I checked the availability on my susy account - unfortunately, there is no access to this data). The solution to such a situation may be for the authors to include such information in the manuscript, nevertheless I performed the reviews trusting that the experiment was properly designed - a critical note.

2.       The references should be and numbered, and there should be numerical references in the text of the manuscript. The citations used by the authors do not comply with the guidelines of the journal

3.        In the text, the authors often forget about proper citation. There are quite a few such shortcomings in the text - for example, line 422 - Zira -> Zira et al. (2022)

4.        line 320 - Authors citing tables from another paper should emphasize whose paper it is i.e. Weidema et al., 2013

5.       Authors should try to use description in scientific language, there are places that should be corrected e.g. Line 96 “Appendix A in the 96 Supplementary Material gives more detail.”  -critical note

6.       in the description of tables Authors at the end of the sentence should not put . (a table is a continuation) e.g. „Table 1. Performance KPI data from the PIC internal benchmark and the prominent industry benchmark, recorded in 2021. Means ± standard deviations”

7.       The authors at line 303 confirm the use of ANOVA. Were tests of homogeneity of variance and normal distribution performed? If so, which ones? There is no information regarding in which program the analyses were performed.

Reviewer 2 Report

Comments and Suggestions for Authors

Sustainability issues, which increase pressure on producers to be more efficient. Sustainable development can address risks and create opportunities to improve efficiency and reduce negative impacts. The authors applied two life cycle assessment (LCA) studies to quantify this. It is also concluded that the selection of production and reproductive traits by ordinary pig breeding provides positive environmental results. I think that all the data in the present study cannot enough support their hypothesis.

1. From the point of view of the research object of the full text, whether it is appropriate to take pigs as the topic.

2. Authors conclude that commercial pig breeding delivers positive environmental outcomes as a result of its selection for production.In fact, pigs are monogastric animals, while sheep and cattle are ruminants. Please explain whether the data from sheep and cattle are sufficient to confirm the author's conjecture

3. The sample size of The table 2 related Reported reductions of GHG emission from livestock production systems, due to genetic selection 's data and the species used for the statistics, especially "sheep", are confusing

4. "Comparison to other studies" in the discussion section is too long, and it is suggested to reorganize the language and rewrite this part.

5. The model developed here can be (and will be) used to assess such trends in other regions than North America and can also be adapted to cater for other production systems  such as nursery-finisher (rather than farrow-to-finish as in the current study) or, in a different dimension, Parma or Ibérico (Italy, Spain). The data in this article are not enough to support this prediction, please explain

Comments on the Quality of English Language

Minor editing of English language required

Round 2

Reviewer 2 Report

Comments and Suggestions for Authors

This form is OK

Comments on the Quality of English Language

none